# Mannose-Binding Lectin Deposition in Membranous Nephropathy and Differentiation of Primary from Secondary Forms

**DOI:** 10.3390/ijms25147659

**Published:** 2024-07-12

**Authors:** Irina Zdravkova, Eduard Tilkiyan, Desislava Bozhkova

**Affiliations:** 1Department of Propaedeutics of Internal Diseases, Medical Faculty, Medical University of Plovdiv, 4000 Plovdiv, Bulgaria; 2Nephrology Clinic, University Hospital “Kaspela”, 4000 Plovdiv, Bulgaria; eet64@yahoo.com; 3Second Department of Internal Diseases, Section “Nephrology”, Medical Faculty, Medical University of Plovdiv, 4000 Plovdiv, Bulgaria; 4Department of General and Clinical Pathology, Faculty of Medicine, Medical University of Plovdiv, 4000 Plovdiv, Bulgaria; desislava_lapteva@abv.bg; 5Department of General and Clinical pathology, Kaspela University Hospital, 4000 Plovdiv, Bulgaria

**Keywords:** lectin–complement pathway, IgG4, APLA2R, deposition, differential diagnosis

## Abstract

The differentiation between primary and secondary forms of membranous nephropathy (MN) is a cornerstone that is necessary for adequate decision making regarding the treatment options and behavior of each specific case. Kidney biopsy and antibody results can be controversial, and a unique biomarker has still not been found. Background and Objectives: We investigated the lack of mannose-binding lectin (MBL) deposition in patients with secondary MNs (sMNs) with the presence of IgG4 deposition in relation to the presence of MBL deposition in patients with primary MNs (pMNs). We also established a connection between the stage of MN and MBL deposition. Materials and Methods: Materials from 72 renal biopsies with proven MN were used for immunohistochemistry staining (IHC) for the phospholipase A2 receptor (PLA2R), immunoglobulin subtype IgG4, and MBL. Patients were separated into one of the following three groups: primary MN (pMN), idiopathic MN (iMN), and secondary MN (sMN). Serum antibodies for PLA2R and thrombospondin type-I-domain-containing 7A (THSD7A) were also used for the precise evaluation of the type of MN, as well as for detecting positivity for PLA2R using IHC. Which stage of MN was present in relation to the deposition of MBL was evaluated. Results: In total, 50 patients were positive for IgG4, 34 with pMN, 12 with iMN, and 4 with sMN. A total of 20 patients were positive for MBL, 14 with pMN and 6 with iMN; no MBL deposits were found in patients with sMN. MBL positivity was predominantly present in the first two stages of MN, with a gradual reduction in the later stages. Conclusions: The activation of the lectin–complement pathway occurs in the early stages of the disease and is associated with the deposition of IgG4; IgG4 deposition is present in sMN, but there is no MBL deposition. IgG4 cannot be used for the differentiation of primary from secondary MNs, but the lack of MBL can be used as a marker for sMN in the early stages of the disease.

## 1. Introduction

Membranous nephropathy is a type of glomerulonephritis, and its incidence has increased over the last decade [1,2]; it is one of the main causes of nephrotic syndrome, representing 25% of cases [3,4]. Classically, the literature describes idiopathic and secondary MNs, which, respectively, represent 70% and 30% of cases [5,6,7]; however, with the discovery of PLA2R antibodies, Laurence Beck, Jr. and David Salant propose the following classification of MNs [8]:Primary MN—75–80%.Anti-PLA 2R-associated: Characterized by the presence of PLA 2R antibodies in serum; these antibodies are also found in subepithelial immune deposits, specifically in relation to immunocomplex (autoimmune) disease.Idiopathic: The rest of the cases are accepted as primary. They may be related to a different type of immune injury, antibodies to another podocyte or glomerular antigen, or undiagnosed secondary nephropathy.

Secondary MN—20–25%: Caused by autoimmune diseases, infections, alloimmunization, neoplasms, toxins, and medications.

In sMNs, the sporadic positivity for Anti-PLA2R and THSD7A antibodies is not excluded. Positive antibodies for Anti-PLA2R have been found in sarcoidosis, lupus nephritis, and Hepatitis B [9]. Several cases of THSD7A expression by tumor cells have also been described [3].

In 2019, Sanjeev Sethi et al. discovered new target antigens in MNs called Exostosin 1 (EXT1) and Exostosin 2 (EXT2) [10]; this discovery resulted in a huge technological leap, using the laser microdissection of glomeruli from PLA2R-negative biopsies, followed by the mass spectrometric identification of trypsin-digested proteins. This approach proved extremely fruitful and contributed to the discovery of the following four new antigens: Exostosins 1 and 2 (EXT1/2), neural epidermal growth factor-like 1 protein (NELL-1), semaphorin 3B (Sema3B), and neural cell adhesion molecule 1 (NCAM1). However, this still does not solve the problem of differentiating primary from secondary MNs, due to the fact that not all antibodies against these antigens are detected in serum, and a large proportion of them are positive in sMNs.

The detection of subepithelial IgG deposits along the capillary loops is characteristic of both primary and secondary MNs. The predominant IgG subclass in pMNs is IgG4, and positive staining for IgG1, IgG3, IgA, or IgM, or significant luminescence in the mesangium, suggests the presence of Systemic Lupus Erythematosus (SLE) or another cause of sMN [11]. Another way to distinguish primary from secondary MNs, although not routinely used, is the presence of staining for PLA2R in immune deposits in a pattern that co-localizes with IgG in APLA2R-associated MNs, as this does not occur in secondary forms [12]. Given the lack of studies on the distribution of IgG subclasses in the different stages of MNs, in 2013, Huang et al. [13] analyzed 157 biopsies with MN with an adequate amount of material for light, immunofluorescence, and electron microscopy. Of these 157 biopsies, 114 were from pMN patients and 43 were from sMN patients. The authors evaluated the intensity in the staining of IgG subclasses and the predominance of the subclasses in the different stages of membranous glomerulonephritis. In pMNs (76%), the IgG4 subclass is the dominant one. Of interest is the finding that, in early-stage (stage I) pMNs, the IgG1 subclass (64%) is the dominant one, and in the remaining later-stage biopsies, the IgG4 subclass is the dominant form. In secondary forms of MN, IgG4 deposition is observed at 49%, and IgG1 deposition is found at 60%. These data indicate that, in early-stage MNs, the antibody response is different from that in the later stages; this suggests that, in the early stages, antigens other than PLA2R have an important role, or that there is an IgG subclass transition in the antibody response, with IgG4 being the dominant subclass of late immunoglobulin.

In 2010, Segawa et al. conducted a study that aimed to elucidate the association be-tween the subclass of deposited immunoglobulins and the complement pathway in patients with MN. They found that the intensity of both factor B and MBL is higher in patients with diffuse MNs. This is the first report on complement activation via a classical pathway that investigated patients with segmental MNs associated with IgG1 and IgG3, as well as complement activation via an alternative and lectin pathway in patients with global MNs associated with IgG2 and IgG4 deposits [14].

In 2018, Norifumi Hayashi et al. also found the co-localization of MBL and IgG4 deposits, along with the predominance of MBL deposits in patients with APLA2R- and THSD7A-associated MNs (55% vs. 20%); the latter group were also MN patients, but no positive antigens were identified. In conclusion, they found that APLA2R- and THSD7A-positive MNs were strongly associated with the activation of the lectin pathway [15]. In December 2020, George Haddad et al. published a study demonstrating that the impaired glycosylation of IgG4 causes the activation of the lectin complement pathway in patients with Anti-PLA2R1-associated membranous nephropathy [16]. Given that C1q is not detected in patients with pMN, and that the presence of MBL co-localized with APLA2R is increasingly described, it is assumed that the activation of complements in MN, which is a proven fact, occurs not via the classical pathway but via the lectin pathway. The authors related this to the data of Malhotra et al., who found that IgG autoantibodies lacking the terminal galactose residue at position Asn 297 of the constant fragment (Fc) were able to bind and activate MBL [17].

Our goal in this study was to prove that the activation of the lectin–complement pathway plays a significant role in the pathogenesis of the disease, as well as in the differentiation of primary from secondary forms of the disease.

## 2. Results

We present the data of 72 patients with membranous nephropathy, separated into three groups in accordance with their immunological and pathomorphological findings, as follows: primary MNs (pMNs), idiopathic MNs (iMNs), and secondary MNs (sMNs). The data relating to the frequency and type of deposition are summarized in Table 1. Cross-tables and Chi-square tests were used for statistical comparisons between MN types and the frequency and type of deposits.

### 2.1. Data of the Frequency and Type of Deposition

#### 2.1.1. Triple-Positive Cases (12 Patients)

Triple-positive cases (IgG4 +, PLA2R +, and MBL +) were present only in patients with primary MNs.They constituted 27% of the pMN group and 0% of the other two groups, with a significant difference (*p* = 0.013) (Figure 1). Immunohistochemistry of a triple positive patient is shown in in Figure 2.

#### 2.1.2. Double-Positive Cases (27 Patients)

In total, double-positive cases constituted 47% of the pMN group and 33% of the iMN group and were not found in the sMN group (0%). The difference between the first two groups and the sMN one was statistically significant (*p* < 0.001).Patients double-positive for PLA2R/IgG4 constituted 43% of the pMN group and were not found in the other two groups (0%); the difference was significant at *p* < 0.001.Eight patients were double-positive for IgG4/MBL, of which six were in the iMN group (33%) and two were in the pMN group (4%), with a significant difference of 29% (*p* = 0.005). The iMN group also showed a significant difference from the sMN group, in which no double-positive patients for IgG4/MBL were detected (*p* = 0.003). Patients double-positive for PLA2R/MBL were not detected in the studied groups (Figure 3). Immunohistochemistry of a double-positive patient is shown in Figure 4.

#### 2.1.3. Positive Cases for One Indicator (22 Patients)

Cases positive for one indicator constituted 26% of the pMN group, 33% of the iMN group, and 44% of the sMN group; however, the difference between the types of MN did not reach statistical significance (*p* = 0.425). In the pMN group, 24% of positive cases for PLA2R were found with a significant difference at *p* < 0.001; no cases were observed in the other two groups (0%). IgG4-positive patients constituted 2% of the pMN group, 33% of the iMN group, and 44% of the sMN group, with a significant difference between the pMN group and the other two groups (*p* < 0.001). No MBL-only-positive patients were identified in any of the groups (Figure 5).

#### 2.1.4. Triple-Negative Cases (11 Patients)

Triple-negative cases dominated among sMN patients, constituting 56% of the group. In the iMN group, triple-negative patients constituted 33%, but constituted 0% in the pMN group, with a significant difference of *p* < 0.001 (Figure 6). Immunohistochemistry of a triple-negative patient is shown in Figure 7.

### 2.2. Correlation between Positive APLA2R Antibodies in Serum and the Number of PLA2R Positives Using IHC

Data for both APLA2R in serum and deposition versus PLA2R in IHC were available for 58 of the patients. According to PLA2R (IHC), 36 patients were categorized as being positive and 22 were categorized as being negative. According to serum APLA2R, 35 were categorized as being positive (serum APLA2R > 20 L/min/m^2^) and 24 were categorized as being negative. There were 32 matches between the two methods for positive cases, and there were 19 matches for negative cases. There were discrepancies in seven patients, of whom four were positive for PLA2R (immunohistochemistry) but negative for APLA2R in serum. In contrast, the remaining three were positive for APLA2R in serum and negative for PLA2R in IHC (Table 2).

An ROC (receiver operating characteristic) curve analysis showed an agreement rate of 87.60% between the diagnosis of positive and negative cases, according to PLA2R (IHC) and APLA2R in serum; the area under the curve (AUC) was 0.876 (95% CI: 0.763 to 0.949, *p* < 0.001). The sensitivity was calculated to be 91.42%, and the specificity was calculated to be 82.60% (Table 3).

The ROC curve between PLA2R (IHC) and APLA2R in serum is illustrated in Figure 8.

### 2.3. The Relative Share of IgG4-Positive Patients

The relative proportion of IgG4-positive patients was determined according to the type of MN: pMN, iMN, and sMN. A total of 50 patients were positive for IgG4, constituting 68.50% of the entire group. In the pMN group, 34 (74%) were positive for IgG4; in the iMN group, 12 (67%) were positive; and in the sMN group, 4 (44%) were positive. The relative proportion of IgG4 positives was the highest in the pMN patients and lowest in the sMN group; however, the difference between the types of MN did not reach statistical significance (*p* = 0.216) (Figure 9).

### 2.4. The Relative Share of MBL-Positive Patients

#### 2.4.1. MBL-Positive Patients According to the Type of MN

MBL-positive patients according to the type of MN—pMN, iMN, and sMN—were evaluated; a total of 72 patients had MBL data, and 20 (27%) were positive. In the pMN group, the relative share of positive cases was 31% (14 out of 45); in the iMN group, the positive cases made up 33% (6 out of 18); and no positive cases were found in the sMN group. The difference between the types of MN regarding the relative proportion of positive patients for MBL did not reach statistical significance (*p* = 0.100) (Figure 10).

#### 2.4.2. MBL-Positive Patients According to the Stage of MN

As concerns MBL-positive patients according to the stage of MN, 6 (8%) patients were categorized as stage I, 35 as stage II (49%), 24 (33%) as stage III, and 7 (10%) as stage IV. MBL-positive patients that were categorized according to MN stage accounted for 33% (2/6) of those in stage I, 31% (11/35) of patients in stage II, 25% (6/24) of patients in stage III, and 14% (1/7) in stage IV. The highest relative proportion of patients with positive results for MBL was found in stage I, and a slight decrease was observed with each subsequent stage; however, the difference did not reach statistical significance (*p* = 0.641) (Figure 11).

## 3. Discussion

Patients that were triple-positive for PLA2R/IgG4/MBL were present only in the pMN group, with statistical significance at *p* < 0.05, ensuring an accurate diagnosis.

Double-positive patients were not found in the sMN group, with a statistical significance at *p* < 0.001, again providing us with the ability to differentiate between primary and secondary forms.

PLA2R/IgG4 double-positive patients were present only in the pMN group. There were only two patients that were pMN-positive for only IgG4/MBL who were categorized in the pMN group due to the presence of (+) APLA2R in serum. From the sources described in the literature, there are single cases of positive antibodies in serum, with the absence of positivity in biopsy material, even when processing the material through immunofluorescence [18]. These two cases support the theory of epitope spreading and the later formation of antibodies to the PLA2R [19]; their biopsies were performed before we started using APLA2R routinely. If the result of the IHC is triple (−), these patients should not be included in the general group. Sometimes, errors in the fixation and storage of the paraffin block give negative IHC results; this happens when the material is not stored or fixed in paraffin properly and the antibodies are destroyed. Then, the IHC results are negative. However, in our patients, this should not be the case because they both tested positive for IgG4 and MBL.

Double positivity for IgG4/MBL was mainly found in the iMN patients. There were no patients that were double-positive for IgG4/MBL in the sMN group, with a statistical significance of *p* < 0.01 compared with the pMN and iMN groups. No MBL-only-positive patients were identified in any group.

Patients double-positive for PLA2R/MBL were not found in any of the groups. The absence of cases that were only positive for MBL, and the presence of MBL only in combination with IgG4 (with or without PLA2R), supported the findings from the data, which indicated that IgG4 is required for the activation of the lectin pathway. M. Endo et al. found the deposition of MBL/MASP-1 in patients with IgA nephropathy mainly in younger people and at an early stage of the disease [20]. We did not find the relationship described in the literature between the stage of MN and the deposition of MBL in patients with MNs; there was also a paucity of data on this issue for IgA nephropathy. However, histological changes have been found in IgA nephropathy, namely more severe proteinuria and a faster progression to the end stage of chronic kidney disease in patients positive for MBL [18,21]. Triple-negativity predominates among the sMN group, with a statistical significance at the *p* < 0.001 level.

IgG4 deposition was mostly only present in patients with iMN and sMN, with a predominance in the sMN group. The high percentage of IgG4 positivity in the sMN group and the lack of MBL deposition in the same group indicated that, for this group, complement activation did not proceed via the lectin pathway. These IgG4 are most likely directed against basement-membrane-implanted carcinoma or other antigens, and they are not aberrantly glycosylated.

The deposition of only IgG4 cannot be used to differentiate primary from secondary MNs; however, in combination with a negative MBL deposition, it can be used as a marker for sMN in the early stages of the disease.

Triple negativity excludes the diagnosis of pMN and is possible in one-third of patients with iMN, although its highest predominance is in patients with sMNs.

## 4. Materials and Methods

### 4.1. Materials

The data of 72 patients with membranous nephropathy, ranging in age between 24 and 86 years, are presented; in total, 41 men and 31 women were treated at the Nephrology clinic of the University Hospital “Kaspela” for a period of ten years. In all patients, the diagnosis was confirmed using a puncture kidney biopsy and laboratory tests, including immunological, histopathological, and immunohistochemical tests. All the patients had a biopsy-proven diagnosis of MN and were tested for Anti-PLA2R and thrombospondin antibodies.

### 4.2. Methods

#### 4.2.1. Immunohistochemical Testing

Immunohistochemical testing was performed in accordance with the manufacturer’s standard protocols. The following antibodies were used: Recombinant Anti-PLA2R antibodies [EPR20483] (ab211573), Anti-IgG4 antibodies (ab232869), and Anti-Mannan-Binding Lectin/MBL antibodies [3B6] (ab23457). All were from the company “Abcam PLC” 152 Grove Street, Waltham, MA 02453, USA.

Serial sections 4 μm thick were prepared from paraffin blocks and mounted on adhesive slides. The sections were deparaffinized and were rehydrated in alcohols of decreasing concentration. Washing was carried out with Bond TM Wash Solution from the company Medical Technology Engineering LTD, Sofia, Bulgaria according to its instructions for use. Prior to performing the immunohistochemical reaction, heat-mediated antigen retrieval was performed by incubating the sample in Bond TM Epitope Retrieval Solution 1 and 2 with a pH 9.0 buffer.

Serial sections from each of the studied cases were tested for the antibodies used. A positive and negative control were prepared for each run of antibody assays. The positive control was chosen according to the manufacturer’s instructions; namely, for Anti-PLA2R, peri-tumoral glomeruli around clear cell carcinomas were used; for Anti-IgG4, prostate carcinomas were used; and for Anti-MBL, hepatocellular carcinomas were used.

Immunohistochemistry was performed according to the manufacturer’s instructions using the Bond Polymer Refine Detection Kit imaging system from the company Medical Technology Engineering LTD, Sofia, Bulgaria.

The negative control for each antibody was prepared using a standard immunohistochemical procedure without instilling the test antibody. The antibodies used during IHC staining were diluted; the dilution for the Anti-PLA2R antibody was 1/2000; for the Anti-Mannan-Binding Lectin/MBL antibody, the dilution factor was 1/250; and for the Anti-IgG4 antibody, the dilution factor was 1/1000.

An Olympus light microscope, No. OD82685, provided by Plovdiv Medical University was used. In each case, at least 5 fields were selected in the serial sections and were observed at 400× magnification (eyepiece: ×10; objective: ×40).

Interpretation of Anti-PLA2R Antibody Staining Results: Two false-positive patterns could be observed. The first was characterized by the presence of a weak linear expression localized to the outer surface of the glomerular loop, which was observed in both the normal kidney and the negative internal controls. In the second pattern, Anti-PLA2R showed a more intense “smudge” staining in the Baumann space, possibly due to the presence of normally expressed PLA2R protein on the podocyte membrane. In contrast to the above expression, the dot-like subepithelial granular pattern was accepted as the only true positive result. The degree of staining was also scored, as follows: (1+) weak expression, (2+) moderate expression, and (3+) strong expression (Figure 12).

The lack of IgG4 IHC expression in the normal kidney represented a complete lack of staining in the negative cases and a similar “dot-like” pattern in the positive cases, making interpretation easier.

Using a similar method, Anti-MBL expression was explored. A complete absence of staining in the negative cases was found, and a “dot-like” pattern was observed in the positive cases.

In cases with advanced disease or extensive segmental sclerosis, IHC positivity was limited to the area of the glomeruli that had not undergone fibrosis, representing a possible cause of false-negative results.

#### 4.2.2. Statistical Analysis

Most of the data were measured on a dichotomous (There is/Yes–There is not/No), nominal, or ordinal scale. These values are presented as both numbers and percentages, and the following methods were used to establish statistically significant trends: Fisher’s exact test for dichotomous quantities and a Chi-square test in the presence of more than two categories. The results are illustrated with bar charts and line charts.

All the statistical analyses were performed at an allowable error level of alpha = 5% (*p* < 0.05). The results were graded according to statistical significance as follows: *—*p* < 0.05; **—*p* < 0.01; ***—*p* < 0.001. The statistical programs IBM SPSS, version 27 (2020); Minitab, version 19 (2020); and MedCalc, version 20.008 (2021) were used for data analysis.

## 5. Conclusions

Our data proved that the activation of the lectin–complement pathway occurs in the early stages of the disease and is associated with the deposition of IgG4. For the first time, MBL deposition in kidney tissue was investigated using IHC and was found to play a role in disease pathogenesis, not only in antigen-associated MNs but also in iMNs.

As has been observed, none of these markers used separately can enable a 100% differentiation between secondary and primary forms. Only combining all three of them together, especially triple positivity, triple negativity, and a lack of MBL in sMNs, can give us the confidence to initiate pathogenetic treatment or to continue the search for a diagnosis associated with sMN. We believe that the routine use of these three markers through immunofluorescence could become very helpful in the future. If the data that we presented are confirmed by other researchers, this will open up new treatment possibilities through blocking the lectin–complement pathway.

## Figures and Tables

**Figure 1 ijms-25-07659-f001:**
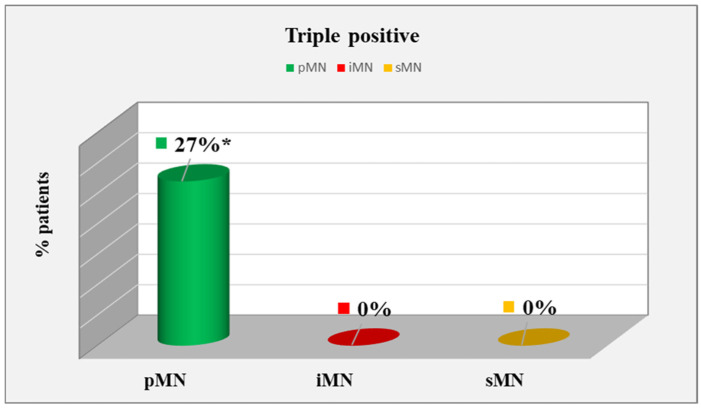
Distribution of triple-positive patients. *—statistically significant difference at *p* < 0.05.

**Figure 2 ijms-25-07659-f002:**
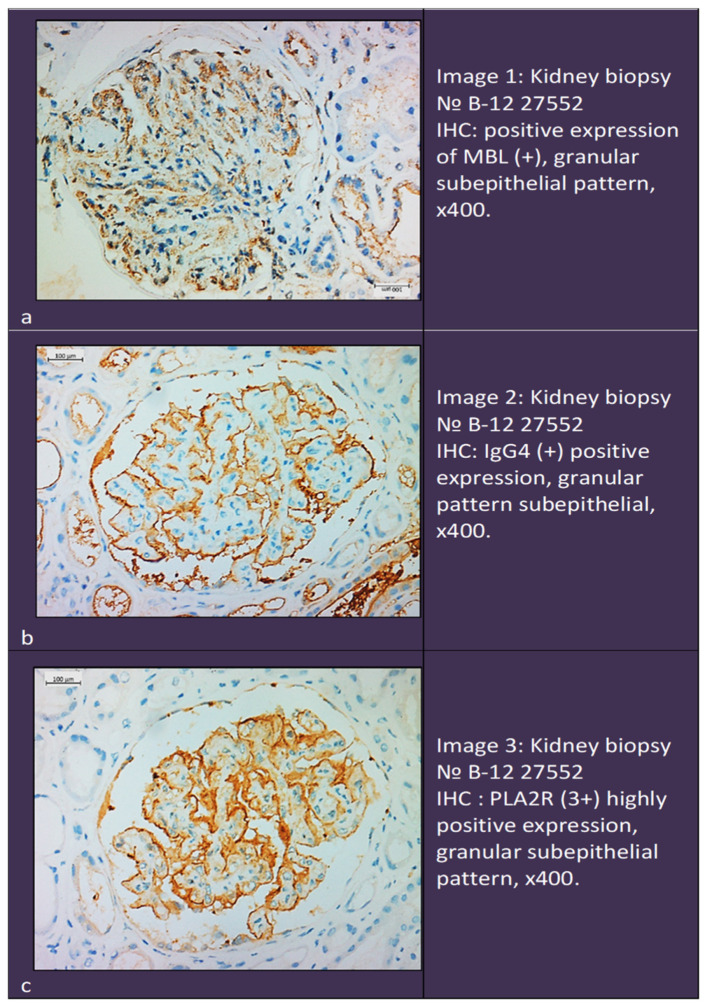
Immunohistochemistry in a triple-positive patient. (**a**). Positive expression of MBL. (**b**). Positive expression of IgG4. (**c**). Positive expression of PLA2R.

**Figure 3 ijms-25-07659-f003:**
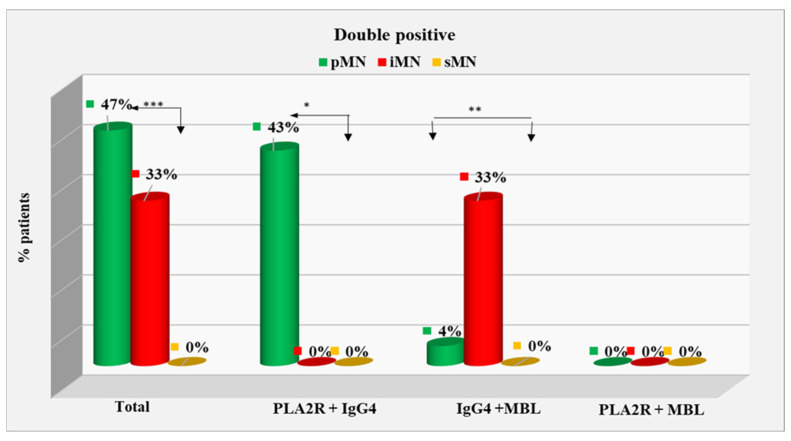
Distribution of double-positive patients. *—statistically significant difference at *p* < 0.05; **—statistically significant difference at *p* < 0.01; ***—statistically significant difference at *p* < 0.001.

**Figure 4 ijms-25-07659-f004:**
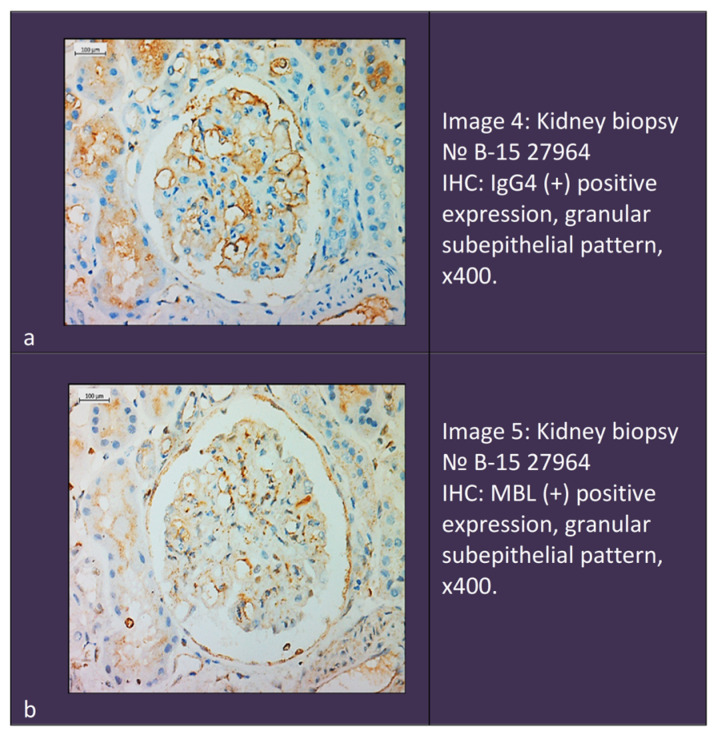
Immunohistochemistry in a double-positive patient. (**a**). Positive expression of IgG4. (**b**). Positive expression of MBL.

**Figure 5 ijms-25-07659-f005:**
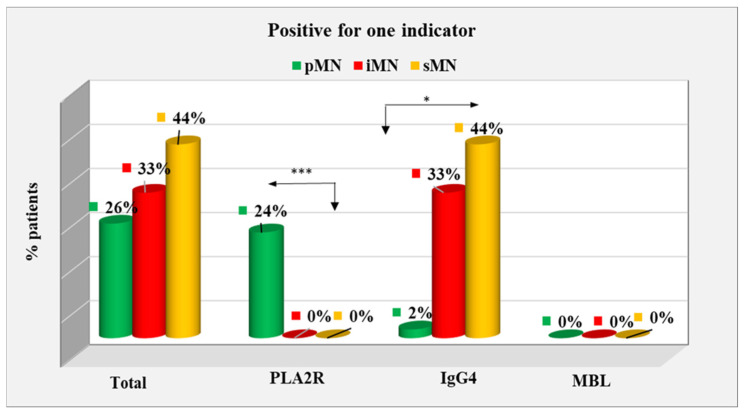
Distribution of patients with positive data for one indicator among types of MN. *—statistically significant difference at *p* < 0.05; ***—statistically significant difference at *p* < 0.001.

**Figure 6 ijms-25-07659-f006:**
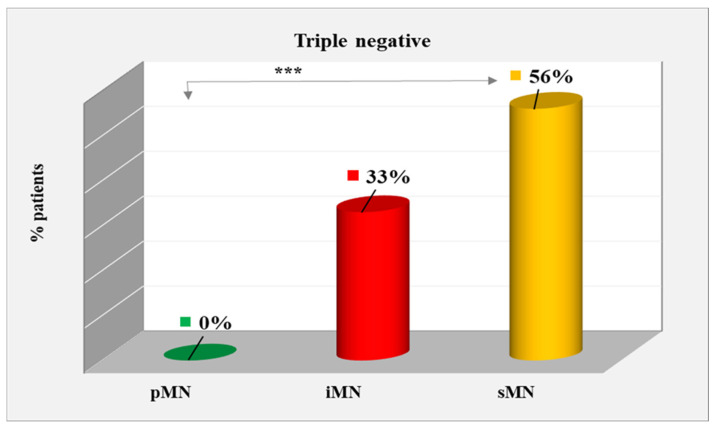
Distribution of patients with triple-negative indicators among types of MN. ***—statistically significant difference at *p* < 0.001.

**Figure 7 ijms-25-07659-f007:**
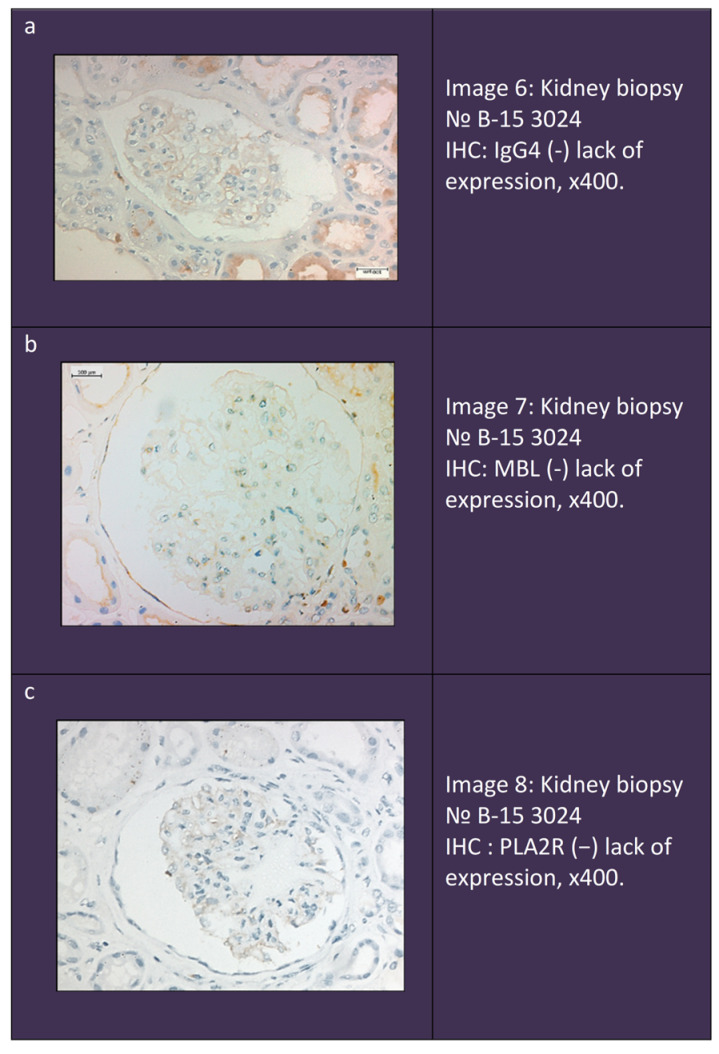
Immunohistochemistry in a triple-negative patient. (**a**). Lack ofIgG4 expression. (**b**). Lack of MBL expression. (**c**). Lack of PLA2R expression.

**Figure 8 ijms-25-07659-f008:**
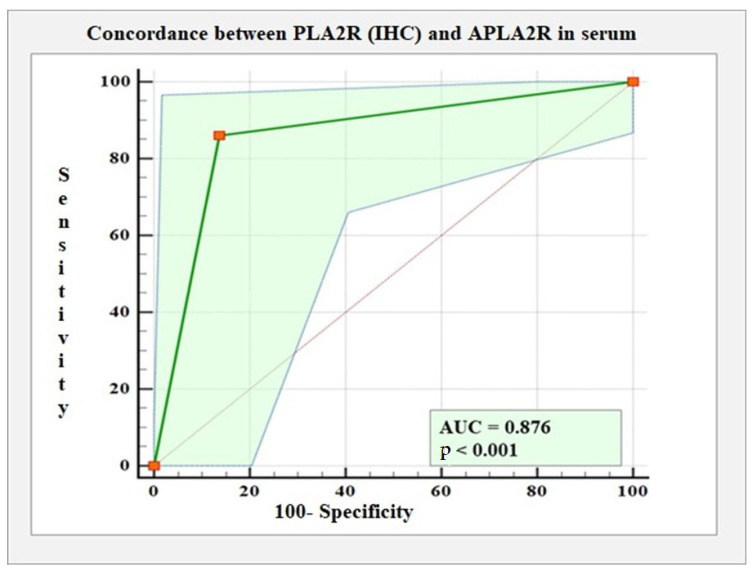
ROC curve between PLA2R (IHC) and APLA2R positivity in serum.

**Figure 9 ijms-25-07659-f009:**
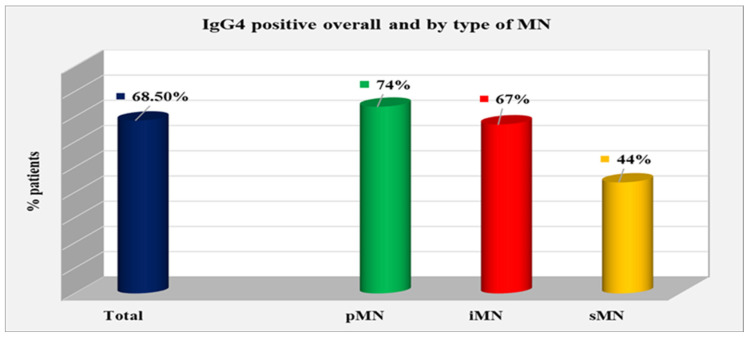
Distribution of IgG4-positive patients overall and according to MN type.

**Figure 10 ijms-25-07659-f010:**
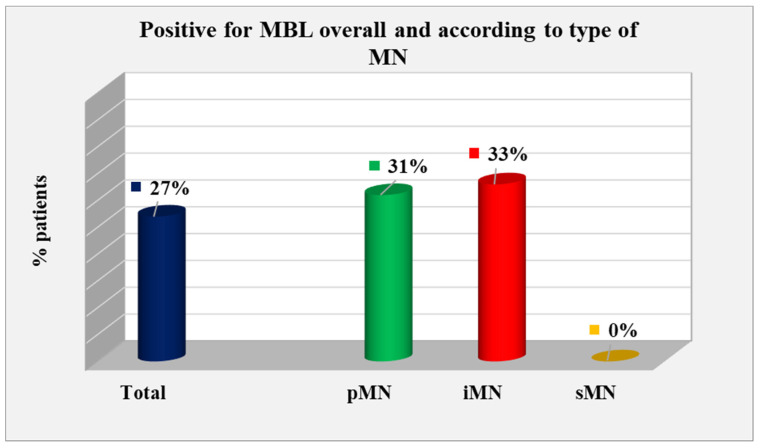
Distribution of MBL-positive patients overall and according to MN type.

**Figure 11 ijms-25-07659-f011:**
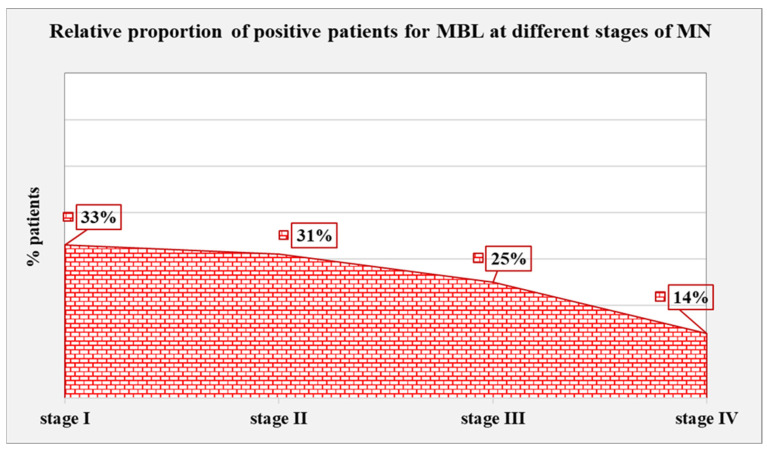
Relative share of positive patients for MBL at different stages of MN.

**Figure 12 ijms-25-07659-f012:**
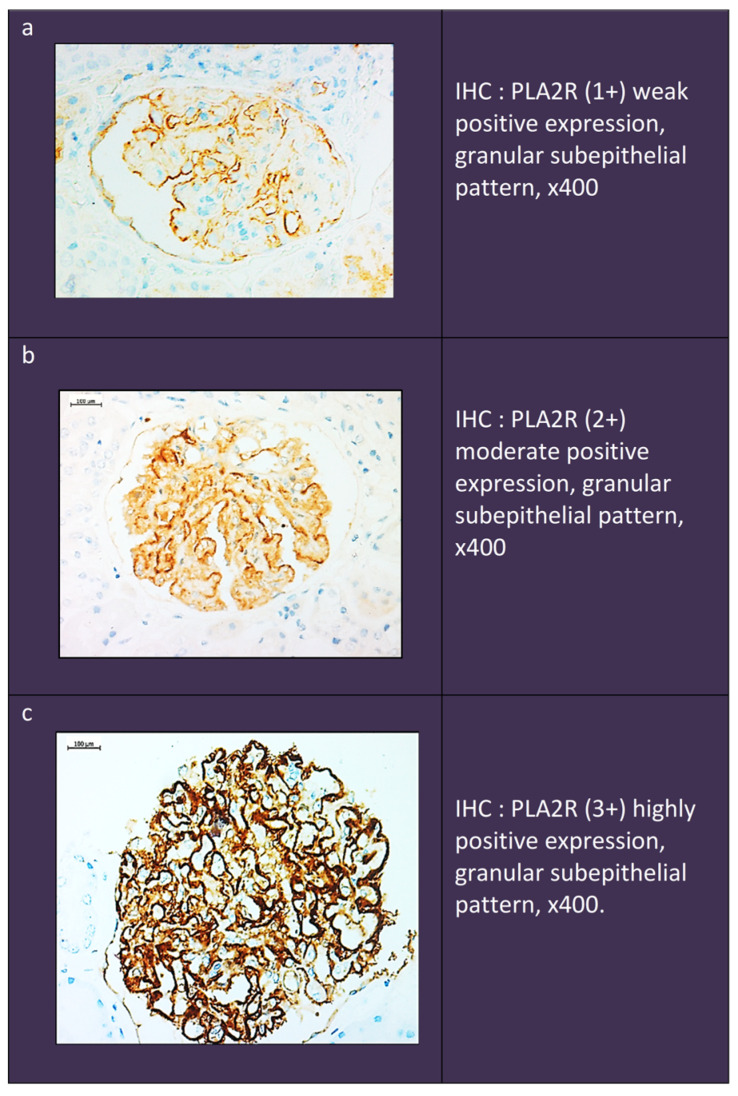
Microscopic images with their scoring of PLA2R expression. (**a**). Weak positive PLA2R expression. (**b**). Moderate positive PLA2R expression. (**c**). Highly positive PLA2R expression.

**Table 1 ijms-25-07659-t001:** Frequency of IgG4, APLA2R, and MBL deposits according to the type of MN.

Value	pMN (n = 45)	iMN (n = 18)	sMN (n = 9)	p
Triple-positive PLA2R, IgG4, and MBL	12 (27%)	0 (0%)	0 (0%)	0.013 *
Double-positive	21 (47%)	6 (33%)	0 (0%)	0.000 ***
o PLA2R and IgG4	19 (43%)	0 (0%)	0 (0%)	0.000 ***
o PLA2R and MBL	0 (0%)	0 (0%)	0 (0%)	NA
o IgG4 and MBL	2 (4%)	6 (33%)!	0 (0%)	0.005 ** 0.003 **
Positive for indicator	12 (26%)	6 (33%)!	4 (44%)	0.425
o PLA2R	11 (24%)	0 (0%)	0 (0%)	0.000 ***
o IgG4	1 (2%)	6 (33%)	4 (44%)	0.000 ***
o MBL	0 (0%)	0 (0%)	0 (0%)	NA
Triple-negative	0 (0%)	6 (33%)	5 (56%)	0.000 ***

***—statistical significance at *p* < 0.05; **—statistical significance at *p* < 0.01; ***—statistical significance at *p* < 0.001; !—significantly higher relative share compared to pMN group (*p* = 0.005); significantly higher relative share compared to sMN group (*p* = 0.003).

**Table 2 ijms-25-07659-t002:** Crosstab between deposition against PLA2R on immunohistochemistry and APLA2R in serum.

pMN	APLA2R in Serum		Total
	Positive (+)	Negative (−)	
Yes	36	4	40
No	0	19	19
Total	36	23	59

**Table 3 ijms-25-07659-t003:** Results of ROC curve analysis for degree of concordance between PLA2R (IHC) and APLA2R in serum.

	AUC (95% CI)	SE	p	Sensitivity (95% CI)	Specificity (95% CI)
APLA2R in serum	0.876 (0.763 up to 0.948)	0.04	0.000	91.42% (76.94% up to 98.19%)	82.60% (61.21% up to 95.04%)

AUC—area under the curve; SE—standard error.

## Data Availability

The data presented in this study are available on request from the corresponding author. The data are not publicly available due to national legal restrictions.

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
