# Peer review of "Mannose-Binding Lectin Deposition in Membranous Nephropathy and Differentiation of Primary from Secondary Forms"

_ijms, 2024, doi:10.3390/ijms25147659_

Round 1

Reviewer 1 Report

Comments and Suggestions for Authors

The manuscript aims to prove the existence of some new biomarkers for the diagnosis and differentiation of primary and secondary membranous nephropathy.

The major method used is immunohistochemistry, however, no microscopy pictures/figures were included in the manuscript to prove the conclusions of the study. Some representative microscopy photos of the analyzed markers should definitely be included, otherwise there is no proof to the claimed scientific conclusions.

Also the English language must be improved overall. The formulation 'lectin complement' should be replaced with 'lectin complement pathway" in the manuscript, and also in the list of keywords.

Comments on the Quality of English Language

English language must be improved as there are many syntax and grammar errors in the manuscript.

Author Response

Comments 1: The major method used is immunohistochemistry, however, no microscopy pictures/figures were included in the manuscript to prove the conclusions of the study. Some representative microscopy photos of the analyzed markers should definitely be included, otherwise there is no proof to the claimed scientific conclusions.

Response 1: Microscopy pictures/figures are included in the manuscript to prove the conclusions of the study. I agree, that otherwise there is no proof to the claimed scientific conclusions and it made the article better!

Comments 2: Also the English language must be improved overall. The formulation 'lectin complement' should be replaced with 'lectin complement pathway" in the manuscript, and also in the list of keywords.

Response 1: The formulation 'lectin complement' is replaced with 'lectin complement pathway" in the manuscript. The article underwent English Language Editing, and some changes in the tables were made due to MDPI requirements.

Reviewer 2 Report

Comments and Suggestions for Authors

Dear Authors,

thank you for preparing this interesting manuscript describing the promising biomarkers of differentiation of primary and secondary forms of membranous nephropathy. I strongly believe such research is necessary and can help in the development of new diagnostic procedures and effective therapies.

However, I have some suggestions for the Authors on how to improve this article:

1. Firstly, I feel there is a slight lack of consequence in this article as the title and the introduction describe only the MBL depletion as the biomarker for distinguishing MN, whereas in the conclusions the Authors clearly write that only by investigating the correlation between IgG4, APLA2R and MBL it could be presumed which stage of MN has occurred. It would be good to smooth out the way these parts are described (title, introduction, conclusions).

2. In the title, I suggest using the full name "mannose-binding lectin" instead of the abbreviation MBL.

3. At the end of the introduction, the Authors should add information on what was their goal in this research.

4. Please consider reorganizing the order of the chapters: first the materials and methods part, and next the results part.

5. Please sort the materials and methods section into subsections. Now some methods are mixed.

6. Please add information on the concentration of antibodies applied during IHC staining.

7. Please correct the entry into English in the table 1 (first column) and table 3 (second column).

8. Please consider adding the information on how many patients were in each group in the subtitle of each chapter 3.1.1. Triple positive cases (12 patients), 3.1.2. Double-positive cases (27 patients), 3.1.3.positive cases for one indicator (22 patients), 3.1.4 Triple-negative cases (11 patients).

9. Line 126, please correct:
"In total, double-positive cases constitute..."

10. Line 129, please double check the number as I believe it should be 43% not 47%?
"Double-positive patients for PLA2R/IgG4 constitute 47% of the pMN group..."

11. Line 134, please add a new description:
"Double-positive patients for PLA2R/MBL were not detected in the studied group"

12. Please add information on the concentration of antibodies used during IHC staining

13. Please present exemplary microscopic images with the scoring of each biomarker: (1+) weak expression, (2+) moderate expression and (3+) strong expression.

14. Some abbreviations need explaining (line 24 - THSD7A, line 68 - SLE, line 243 - CKD)

15. Please unify in the whole text the writing of some abbreviations (e.g., line 51 Anti-PLA 2R and line 52 Anti-PLA2R)

16. Please remove some unnecessary dashes (lines 97, 113, 136, 142, 149, 161, 164, 166, 167, 274, 279, 286

17. Please add dashes in "triple-negative" (fig. 4, line 159, 318) IgG4-positive (line 194), MBL-positive (line 204)

18. Please correct the abbreviation IHC (lines 293, 298

With kind regards

Author Response

Comments 1: Firstly, I feel there is a slight lack of consequence in this article as the title and the introduction describe only the MBL depletion as the biomarker for distinguishing MN, whereas in the conclusions the Authors clearly write that only by investigating the correlation between IgG4, APLA2R and MBL it could be presumed which stage of MN has occurred. It would be good to smooth out the way these parts are described (title, introduction, conclusions).

Response 1: I agree, this is why I decided to change the title.

Comments 2: In the title, I suggest using the full name "mannose-binding lectin" instead of the abbreviation MBL.

Response 2: Thank you, corrected.

Comments 3: At the end of the introduction, the Authors should add information on what was their goal in this research.

Response 3: Thank you, added.

Comments 4: Please consider reorganizing the order of the chapters: first the materials and methods part, and next the results part.

Response 4: At the begging I used the template from the IJMS, and I though this is the way they want it. I reorganized the order and it is better now.

Comments 5: Please sort the materials and methods section into subsections. Now some methods are mixed.

Response 5: Corrected. 

Comments 6: Please add information on the concentration of antibodies applied during IHC staining.

Response 6: Information added.

Comments 7: Please correct the entry into English in the table 1 (first column) and table 3 (second column).

Response 7: The entry into English in the table 1 is corrected, but the outlook of the table is changed also, due to MDPI requirements /no font, no Bolt and Word fomate/.

Comments 8:  Please consider adding the information on how many patients were in each group in the subtitle of each chapter 3.1.1. Triple positive cases (12 patients), 3.1.2. Double-positive cases (27 patients), 3.1.3. positive cases for one indicator (22 patients), 3.1.4 Triple-negative cases (11 patients).

Response 8: Information added.

Comments 9: Line 126, please correct:
"In total, double-positive cases constitute..."

Response 9: Corrected.

Comments 10: Line 129, please double check the number as I believe it should be 43% not 47%?
"Double-positive patients for PLA2R/IgG4 constitute 47% of the pMN group..."

Response 10: Thank you very much, I am sorry for this mistake!

Comments 11:  Line 134, please add a new description:
"Double-positive patients for PLA2R/MBL were not detected in the studied group"

Response 11:  Added.

Comments 12: Please add information on the concentration of antibodies used during IHC staining

Response 12: Information added.

Comments 13: Please present exemplary microscopic images with the scoring of each biomarker: (1+) weak expression, (2+) moderate expression and (3+) strong expression.

Response 13: Microscopic images added, as well as scoring, excuse me if I am not very good in formatting, I hope some specialict can make them smaller, and become bigger after one clik.

Comments 14: Some abbreviations need explaining (line 24 - THSD7A, line 68 - SLE, line 243 - CKD)

Response 14: Full names are given.

Comments 15: Please unify in the whole text the writing of some abbreviations (e.g., line 51 Anti-PLA 2R and line 52 Anti-PLA2R)

Response 15: Abbreviations unified.

Comments 16: Please remove some unnecessary dashes (lines 97, 113, 136, 142, 149, 161, 164, 166, 167, 274, 279, 286

Response 16: Unnecessary dashes removed.

Comments 17:  Please add dashes in "triple-negative" (fig. 4, line 159, 318) IgG4-positive (line 194), MBL-positive (line 204)

Response 17: Thank you, corrected, in the table also.

Comments 18: Please correct the abbreviation IHC (lines 293, 298

Response 18: Corrected.

The article underwent English Language Editing, as well as some corrections from MDPI.

With kind regards,

Irina Zdravkova

Round 2

Reviewer 1 Report

Comments and Suggestions for Authors

The authors have improved the manuscript and have properly addressed the previous criticisms. However, the Tables and the graphs look somehow unprofessional and a different design should be used, also make sure to follow the Journal's instructions and standards.